# Peptide-Based Strategies for Targeted Tumor Treatment and Imaging

**DOI:** 10.3390/pharmaceutics13040481

**Published:** 2021-04-02

**Authors:** Abiodun Ayo, Pirjo Laakkonen

**Affiliations:** 1Translational Cancer Medicine Research Program, Faculty of Medicine, University of Helsinki, 00014 Helsinki, Finland; abiodun.ayo@helsinki.fi; 2Laboratory Animal Center, HiLIFE—Helsinki Institute of Life Science, University of Helsinki, 00014 Helsinki, Finland

**Keywords:** tumor homing peptides, drug delivery, targeted delivery

## Abstract

Cancer is one of the leading causes of death worldwide. The development of cancer-specific diagnostic agents and anticancer toxins would improve patient survival. The current and standard types of medical care for cancer patients, including surgery, radiotherapy, and chemotherapy, are not able to treat all cancers. A new treatment strategy utilizing tumor targeting peptides to selectively deliver drugs or applicable active agents to solid tumors is becoming a promising approach. In this review, we discuss the different tumor-homing peptides discovered through combinatorial library screening, as well as native active peptides. The different structure–function relationship data that have been used to improve the peptide’s activity and conjugation strategies are highlighted.

## 1. Introduction

Peptides are small molecules—often less than 40 amino acids in length—that are derived from natural or synthetic sources. Several synthetic peptides have been identified after combinatorial biopanning of peptide libraries via “one-bead one-compound” (OBOC), phage display, or other screens. In addition, many peptides have been derived from naturally bioactive proteins, such as somatostatin and transferrin. Peptides of synthetic or natural origins are, however, amenable to structural modifications owing to their poor pharmacokinetics. Thus, the target binding sequence motifs have been identified and validated after structure–activity relationship (SAR) profiling using several functional assays and sequence algorithm-based programs, such as BLAST homology search, in silico analysis, and SRMAtlas [1,2,3,4,5]. Given the natural and artificial sources of targeting peptides, it is essential that tumors and/or their associated microenvironment, including vascular cells, extracellular matrix, and immune cells overexpress accessible, specific and functional biomarkers for efficient targeting and imaging.

Several tumor homing peptides have been identified by using phage display technology (Figure 1) [6]. In addition to integrins and other cell surface proteins, many intracellular proteins are often highly expressed on the surface of cancer cells and constitute molecular targets for phage displayed peptides [7,8,9,10]. These peptides can act as carriers to deliver selectively and specifically imaging agents, anticancer toxins, nanoparticles, and/or other applicable active agents to tumors. In addition to the peptide’s binding affinity towards its target (affinity efficiency), it may also show other intrinsic properties, including cytotoxic activity (therapeutic peptides) and/or high permeability (cell/tumor penetrating peptides).

Due to several challenges associated with systemic therapy including non-specificity, low permeability and low retention, off-target toxicities, and in case of brain tumors, inability to cross the blood–brain–barrier (BBB), an efficient and viable strategy that would increase the delivery capacity of anticancer cargoes into the target site without compromising the drug’s efficacy would open many new avenues for targeted drug delivery. It is therefore necessary to improve peptides’ cell penetrating and homing capabilities. It is also noteworthy to address the challenges associated with the complex conjugation chemistry of the peptide-based drug delivery systems.

We review here some of the recent advances in peptide-based delivery systems focusing on peptides for which the target protein is known and that target (i) brain tumors and BBB; (ii) tumor vasculature; (iii) cancer specific signatures involved in invasion and pre-metastatic niche formation; (iv) therapeutic peptides; and (v) cell and tumor penetrating peptides (Table 1). In addition, we discuss the future perspectives towards the development of novel homing peptides.

## 2. Peptides Derived from Combinatorial Screens

Phage display is the most frequently used method of choice for screening tumor targeting peptides. The bacteriophage display peptide libraries on the surface of the phage to allow binding to their potential target molecules. Targets are usually proteins, but may also include carbohydrates, phospholipids, and nucleic acids. For the in vivo phage display, phage library is injected via the tail vein into tumor bearing animals. The target organ is excised, and the bound phage rescued by amplification. The amplified phage is purified and enriched in subsequent cycles of the selection, called biopanning. Finally, enriched clones are selected for sequencing and validation [36] (Figure 1).

### 2.1. Glioma Targeting Peptides

Gliomas represent the most common brain malignancies in adults with unsatisfactory and unmet medical care. Systemic therapy remains challenging due in part to the non-selective and non-specific nature of chemotherapies. The inability of active agents to cross the BBB has further hampered the efficient glioma targeting and overall patient outcome. Therefore, any molecule that could circumvent the BBB to gain access to brain tumors would be a viable delivery tool for the diagnosis and treatment of gliomas (Figure 2). Glioma cells and their associated vascular network selectively express tumor specific biomarkers on their cell surfaces. For example, glioma cells and/or the BBB express abundant cell surface receptors, including mammary-derived growth inhibitor/fatty acid binding protein 3 (MDGI/FABP3) [8], transferrin [37,38], insulin [39,40,41,42,43,44], glucose transporters (GLUT-1) [45,46,47,48], large-neutral amino acid transporter LAT-1 [49,50,51,52,53], and the low-density lipoprotein receptor-related protein 1 (LRP-1) [54,55,56,57,58]. Peptides that specifically recognize these receptors have been utilized to deliver chemotherapy [8,59,60,61], imaging probes [62], siRNA [63,64], DNA [65,66], and nanoparticles [67] to the brain tumors. It is not known how some of these peptides cross the BBB, but it is thought that some of them utilize the receptor-mediated transcytosis [68]. Below, we discuss some of the known and widely studied glioma homing peptides, such as CooP, T12, angiopep-2, and AP.

#### 2.1.1. MDGI/FABP3 Targeting Peptide—CooP

Glioblastoma or grade IV glioma is the most common and aggressive brain tumor in adults [69,70,71]. We previously identified a linear glioblastoma-targeting nonapeptide (CooP: CGLSGLGVA) [8] by using the in vivo phage display [72]. MDGI/FABP3 was identified as the CooP-interacting partner [8]. MDGI is expressed by the glioblastoma cells and their associated vasculature. MDGI expression in human gliomas is grade dependent with the highest expression in glioblastomas. Since its discovery, CooP-peptide has been used to enhance the delivery of nanoparticles and chemotherapy to gliomas and other tumor types overexpressing MDGI [8,67,73].

We reported CooP homing to different intracranial animal models of glioma [8] including the murine astrocyte-derived Hif-1α-deficient (HIFko) [74] glioma. CooP radiolabeled with ^111^In showed in vivo homing to glioblastoma bearing mice in SPECT-CT (Gamma Medica Inc., Northridge, CA, USA) whole-body imaging, but no homing was detected with the radiolabeled scrambled control peptide [8]. A peptide-drug conjugate comprising of a cell penetrating peptide ARF (1–22) [11] incorporated between CooP and an anticancer cargo (chlorambucil) enhanced the cell penetrating and targeting capacity of the drug. Consequently, the drug conjugate showed better anti-glioma efficacy compared to the free drug and prolonged the lifespan of mice bearing invasive brain tumor in our intracranial HIFko glioma model [8].

In addition, the CooP-peptide conjugated to fluorescently labeled nanoparticles accumulated into the MDGI-expressing subcutaneous breast cancer xenografts [73]. For the design, synthetic porous silicon nanoparticles (NP) were fluorescently labeled with the TRITC dye and conjugated to CooP peptide (THCPSi-CooP NPs). Confocal microscopic imaging and quantitative analysis of the histologic tissues revealed approximately 9-fold increase in the targeted NP-derived TRITC signal in tumors compared to the non-targeting NPs (THCPSi-NPs) indicating efficient NP targeting by the CooP [73].

In another study, intravenously administered paclitaxel (PTX) containing PEG–PLA nanoparticles conjugated to CooP (CooP-NP-PTX) showed higher anti-glioma effect in the U87MG-bearing mice compared to the non-targeting control NPs (Taxol, saline, and NP-PTX). More CooP-NPs accumulated in the tumor compared to the unconjugated NPs suggesting specific targeting. In vitro data also supported the efficacy of CooP-targeted therapy in the U87MG glioblastomas [67].

Recently, studies by alanine scan and microscale thermophoresis revealed an improved analogue of the CooP peptide called the A-CooP-K, which showed about 30-fold higher affinity (70 nM K_D_) towards the target MDGI/FABP3 compared to the original CooP peptide with micromolar affinity (2.18 µM K_D_) (Figure 3A). Both CooP and A-CooP-K homed efficiently to patient-derived BT12 glioma xenografts (Figure 3B) [12]. The A-CooP-K also showed excellent cell penetrating capacity that is lacking in the original CooP peptide (Figure 3C). The sequence-function data also showed that Cys^1^, Gly^5^, and Gly^7^ in the CooP sequence are the most essential amino acids for the MDGI/FABP3-CooP interaction [12].

#### 2.1.2. Transferrin Receptor Targeting Peptides—T7/T12

Transferrin receptor (TfR), is a transmembrane glycoprotein involved in cellular iron trafficking after interaction with its native iron-rich ligand, transferrin (Tf) [37,75,76,77]. Of the two TfR subtypes, theTfR2 is abundantly expressed in glioblastomas and their associated vasculature [78]. Using the M13-phage screening on chicken embryo fibroblasts constitutively expressing the human transferrin receptor (hTfR), peptides T7 (HAIYPRH) and T12 (THRPPMWSPVWP) were discovered [13]. Both peptides have been widely used to target several cancer types overexpressing transferrin receptors (TfR1 and TfR2), including liver, prostate, breast, pancreatic cancers, and many others.

T12 showed higher affinity, in nanomolar range, towards the target and appeared more promising than the T7 peptide [13]. However, T12 failed to show enhanced transport of conjugated compounds across the BBB [79]. Meanwhile, T7 peptide was classified as a target unrelated peptide (TUP). Due to a mutation in the Shine–Dalgarno sequence for the gIIp gene of the phage, the M13 phage displaying the HAIYPRH peptide amplified dramatically faster than the other phage leading to the enrichment of this sequence during the biopanning [80]. Since then, it has become evident that these TUPs become enriched in each and every screen during biopanning. The target unrelated enrichment may be due to either selection or propagation. The selection-related TUPs represent peptides that bind to a component of the screening system other than the target, i.e., microtiter wells, streptavidin or bivalent metal ions. The propagation-related TUPs on the other hand are peptides that are coincidentally displayed on a library phage clone that contains an advantageous mutation that allows faster propagation [81,82,83,84]. The enrichment of the selection-related TUPs concerns mainly the in vitro but not the in vivo screens. While the enrichment of the propagation-related TUPs is a concern in all types of screen. To facilitate the identification of TUPs, an online database SAROTUP (Scanner and Reporter of Targeted-Unrelated Peptides) was setup by compiling the known and/or suspected TUP sequences [85] with a recently upgraded version [86]. Unfortunately, the link appears not active but the publications contain information about some of the TUPs. However, despite significant literature on TUPs, several research groups are still working on improving the T7 peptide-based targeted delivery systems with the latest publications during 2020.

#### 2.1.3. LRP Targeting Peptides—Angiopep-2

The low-density lipoprotein receptor related protein 1 (LRP-1) binds multiple ligands and controls the BBB permeability [56,57]. LRP-1 has been shown to be highly expressed in glioblastoma cells and glioblastoma-associated endothelial cells [14,58]. LRP2, or megalin, a multi-ligand receptor like LRP1, is expressed by many resorptive epithelia, thus indicating a role in endocytosis and transport [57]. In addition, it mediates the transport of several macromolecules such as lactoferrin and melanotransferrin across the BBB [87,88]. Demeule and colleagues developed shortened peptide variants from the active LRP-binding domain (Kunitz-type) of aprotinin and named these angiopeps (angiopep-1, -2, -5, and -7) [89]. The BBB permeability of the peptides was tested by using an in vitro BBB model and in situ mouse brain perfusion. A 19-amino acid aprotinin peptide (angiopep-2; TFFYGGSRGKRNNFKTEEY) showed the highest permeability and LRP-mediated transcytosis compared to the other aprotinin-derived peptides and the original aprotinin [14,89]. Since then, angiopep-2 have been successfully used to deliver therapeutic/imaging agents mostly to primary and secondary brain tumors. Consequently, a peptide-drug conjugate consisting of angiopep-2 and paclitaxel drug called ANG1005 or also known as GRN1005, has been developed and used to target both primary brain tumors and brain metastases in pre-clinical studies [90,91,92] and clinical trials [93,94,95,96].

Angiopep-2 was also used for the delivery of nanoparticles into the brain where they accumulated in the neurons in different brain areas [97]. In addition, a polyethylene glycol (PEG) was incorporated into an “ultrasmall superparamagnetic iron oxide nanoparticles (USPIONs)” to improve their retention time in circulation, followed by conjugation to angiopep-2 (ANG/PEG-USPIONs), for enhanced delivery. The dual targeting nanoprobe showed enhanced signal and was stable up to 4 h post-injection upon in vivo MR imaging of glioblastoma bearing mice while the non-targeting controls (PEG-USPION and Gd-DTPA) showed lower signal intensities. Permeability of the nanoprobes across the in vitro BBB comprising of apically layered endothelial cells (BCECs) and basolaterally layered glioblastoma cells (U87MG) were tested. Confocal laser microscopy and flow cytometry assays showed enhanced permeability and U87MG cellular uptake of the targeting nanoprobe (ANG/PEG-USPION; 24.3 ± 4.0%) compared to non-targeting control (PEG-USPIONs; 5.5 ± 1.3%) [98].

#### 2.1.4. Interleukin-4 Receptor Targeting Peptide—AP

Atherosclerotic plaque (AP) homing nonapeptide (CRKRLDRNC) was identified after cycles of ex vivo phage display screening on human atheroma tissue and further validated for homing to the atherosclerotic plaques in an in vivo murine model. A homology sequence search identified the AP binding sequence ^84^KRLDRN^89^ motif corresponding to the Interleukin-4 (IL-4) [15]. Its receptor, IL-4R, is abundantly expressed by glioma cells [99] thereby presenting the CRKRLDRNC as a promising tool for glioma targeting. Park et al. showed promising results with the AP peptide conjugated to doxorubicin containing microbubbles and nanoliposomes as a proof-of-concept for glioma targeted imaging and therapy using the U87MG in vitro models [100]. AP peptide conjugated to doxorubicin containing liposomes showed also enhanced antitumor activity in in vivo model of human lung cancer [101].

### 2.2. Tumor Vasculature Targeting Peptides

Angiogenesis or formation of new blood vessels is one of the hallmarks of cancer [102]. Nutrients needed for tumor growth are provided through the blood vessels thereby implicating angiogenic blood vessels or neovasculature as a viable therapeutic target. Tumor blood vessels express unique set of proteins that are different from the vasculature of the corresponding normal organ but may be expressed by the angiogenic vessels in non-pathogenic conditions [103]. Neovasculature expresses several cell surface targets including various integrin subunits (αvβ3, αvβ5 and αvβ1) [104], matrix metalloproteases and other receptors required for tumor cell adhesion and motility, as well as tumor vessel growth [103]. In vivo phage displayed peptide screens have identified various peptide sequences homing to cell surface receptors expressed on tumor-associated blood vessels. These tripeptide motifs containing sequences RGD (arginine–glycine–aspartic acid), NGR (asparagine–glycine–arginine), and GSL (glycine–serine–leucine) with their corresponding improved variants have been extensively studied. Some of these studies are reviewed below.

#### 2.2.1. First Generation Integrin Targeting Peptides

The two conventional RGD containing peptides, CRGDC and CDCRGDCFC, also known as RGD-4C, recognize tumor associated blood vessels by interacting with various integrin subunits [16,17]. Several native extracellular matrix (ECM) proteins, such as fibronectin [105,106,107,108,109], laminin [110], vitronectin [111], osteopontin [112], tenascin [113], collagens I, IV [114,115], and fibrinogen [116,117,118,119] interact with the integrin receptors including integrin αvβ3 through the RGD domain. The αvβ3 integrin is highly expressed on the tumor cells and angiogenic endothelial cells, and its expression associates with tumor progression and metastasis, tumor angiogenesis and inflammation [120,121,122]. RGD-4C peptide has been widely used to deliver therapeutic agents including doxorubicin (doxo-RGD4C) [18,123], tumor necrosis factor (TNF) [124,125,126], proapoptotic peptide (KLAKLAK)_2_ [127], and other active agents, both in in vitro and in vivo studies [128,129,130]. In a similar fashion, several RGD-targeted radionuclides have been studied as diagnostic agents using either positron emission tomography (PET) [131,132,133,134,135,136,137,138,139,140,141] or single-photon emission computed tomography (SPECT) [142,143,144,145,146,147]. Several modifications have been made to the original RGD containing peptides to increase their tumor penetration and accessibility, either by generating cyclic or improved linear RGD variants [19,148].

#### 2.2.2. Monocyclic and Multicyclic RGD Peptides

To improve the permeability and selective activity of the RGD containing peptides, sequence-function studies by NMR spectroscopy and molecular dynamics simulations were performed on synthetically generated cyclic oligopeptides of the formulated cyclic RGD. The inhibitory activities of cyclic RGD-derived variants or linear peptides were tested against laminin and/or vitronectin fragment P1 in three different cell lines, including HT-1080 fibrosarcoma, HBL-100 mammary epithelia, and A375 melanoma cell lines. Incorporation of either D-Phe (f) or D-Val (v) into the cyclic RGD, i.e., c(RGDfV) or c(RGDFv) increased their inhibitory activity, while replacement of Gly with either D-Ala or L-Ala reduced the inhibitory effect compared to the linear variants, such as GRGDS and RGDFV. The comprehensive conformation analysis presents different active cyclic RGD analogues that have binding orientation close to the native ligands. For example, the introduction of D-Phe (f), D-Val (v) or D-Pro (p) residues to the cyclic RGD is important for the integrin receptor binding efficiency [20,149]. Crystal structure of the extracellular segment of integrin αVβ3 in complex with the RGD ligand has also been resolved [150]. The single amino acid replacement in the RGD sequence may culminate in steric hindrance, thereby abolishing interaction with the αvβ3 integrin [151]. Several similar studies revealed that the cyclic RGD including the cyclic polymeric RGD-containing peptides have the highest binding activities towards integrin targets compared to the linear RGD or the monomeric cyclic variants [20,152,153]. For example, cyclic tetrameric RGD (cRGDfK)_4_ conjugated to Cy5 showed 10-fold higher affinity towards the αvβ3 target (3.87 nmol/L K_D_) compared to the monomeric cRGD-Cy5 (41.70 nmol/L K_D_) [154].

N-methylation of the cRGDfV was shown to increase selectivity and inhibition of the vitronectin binding to the αvβ3 integrin [21]. The c(RGDfN(Me)V) designated as cilengitide (Merck, EMD 121974, Germany) is a widely used αv-integrin (αvβ3, αvβ5) antagonist [155,156] and has been used in over 30 clinical trials to date [157,158,159,160,161,162]. In a Phase II clinical trial for recurrent glioblastoma multiforme (GBM), single-agent cilengitide showed minimal toxicity and modest antitumor activity [163]. Among newly diagnosed, GBM patients, single-arm studies incorporating cilengitide into standard external beam radiotherapy/temozolomide treatment showed encouraging activity with no increased toxicity, and led to a randomized Phase III trial [164,165]. However, cilengitide treatment, when provided in addition to the standard radio-chemotherapy, did not meet the endpoints and did not improve the patient outcomes [166,167], and its development as an anticancer drug was discontinued [167]. The possible reasons for the negative Phase III outcome as well as whether cilengitide deserves a second chance have been discussed [168,169,170].

#### 2.2.3. CD13 Targeting Peptide—NGR

Another very intensively studied peptide originally identified by using phage display is the NGR [16,18,171]. The NGR containing peptide (CNGRC) selectively homes to tumor associated angiogenic blood vessels by binding to the aminopeptidase N (APN), also known as CD13 [17,18,172]. NGR can rapidly undergo asparagine deamination to generate an isoaspartate-glycine-arginine (isoDGR) peptide capable of binding to αvβ3 [173,174,175]. Both the linear and cyclic NGR peptides have been used for targeted delivery of cytokines (TNF-α) [176,177], chemotherapy (doxorubicin) [178,179], radiopharmaceuticals [180,181,182], and pro-apoptotic peptides [127]. Anti-TNF agents are in clinical use for cancer treatment but due to the systemic dose related toxicity, a recombinant NGR–TNF fusion protein was developed. The NGR–TNF showed enhanced efficacy against murine lymphoma and melanoma even at lower doses compared to the TNF [177]. Several chemotherapeutic drugs including doxorubicin, paclitaxel, gemcitabine, cisplatin, and melphalan have been used in preclinical studies in combination with NGR–TNF for targeted therapy against RMA lymphoma and WEHI-164 murine fibrosarcoma tumor bearing mice. Both the in vivo and in vitro data revealed improved antitumor efficacy compared to the TNF [183]. Structure-activity data also revealed that the cyclic CNGRC peptide conjugated to TNF showed 10-fold higher antitumor activity than the linear GNGRG–TNF compound. The cyclization of the NGR peptide through the disulfide bridge between the two cysteine residues increased the stability and targeting efficiency of CNGRC–TNF [184].

#### 2.2.4. p32 Targeting Peptide—LyP-1

LyP-1 (CGNKRTRGC), a cyclic phage-displayed nonapeptide was identified as a tumor lymphatic vessel homing peptide [22]. LyP-1 accumulated in the tumor lymphatics but not the tumor blood vessels of several tumor xenograft models including osteosarcomas, prostate and breast cancers, and metastatic lymph nodes of the vascular endothelial growth factor C (VEGF-C) overexpressing MDA-MB-435 tumors. In addition, LyP-1 recognizes the hypoxic tumor lesions [22]. LyP-1 is a unique peptide due to its tumor penetrating ability and the inbuilt antitumor activity [185]. LyP-1 was internalized and inhibited the MDA-MB-435 tumor cell growth in a dose dependent manner with an IC_50_ of approximately 66 µM. This effect was LyP-1 specific and no effect on cell viability was detected when cells were treated with the control peptides (CRVRTRSGC, CGEKRTRGC) [185]. Later the cell surface associated mitochondrial protein p32/gC1qR was identified as the LyP-1 interacting partner expressed by tumor cells and tumor-associated macrophages that are often incorporated into the walls of tumor lymphatic vessels [7]. p32 is expressed in several tumor types as well as in atherosclerotic lesions [186,187,188,189]. The LyP-1 peptide has been used in several applications for tumor targeted delivery of imaging and therapeutic agents [190]. Timur et al. elucidated the interaction between LyP-1 and p32 by generating alanine scan derivatives of the LyP-1 peptide followed by interaction studies using molecular docking and Molecular Mechanics Poisson–Boltzmann Surface Area (MM–PBSA). The structure-activity relationship data generated by alanine-scan thermodynamics revealed ^3^Asn, ^4^Lys, ^5^Arg, and ^7^Arg as the most essential amino acids responsible for the p32 binding [191]. The shortened variant of LyP-1 (truncated LyP-1; tLyP-1) with the highest tumor penetration activity conferred by its exposed CendR internalizing domain is reviewed under the tumor penetrating peptides.

### 2.3. Peptides Targeting Tumor Invasion and Pre-Metastatic Niche Formation

In vitro phage display has been utilized against several cancer specific receptors implicated in cell migration, proliferation, cell adhesion, remodeling of extracellular matrix (ECM), and pre-metastatic niche formation. Here, we highlight the success of identification of tumor-targeting peptides against purified and isolated proteins involved in tumor malignancy such as cancer stem cell markers, matrix metalloproteinases (MMP2, MMP9), growth receptor bound protein 7 (Grb7), and epidermal growth factor receptor (EGFR). The selected proteins have been shown to promote cancer invasion and metastasis in different in vitro and in vivo models [192].

#### 2.3.1. EGFR Targeting Peptide

EGFR is a crucial driver of tumorigenesis abundantly overexpressed, amplified or mutated in many cancers including lung and breast cancers, and glioblastomas [193,194]. EGFR overexpression or mutation often correlate with drug resistance thereby presenting EGFR as a possible target in many malignant tumors [195]. For the identification of EGFR binding peptides, T7 phage displayed peptide library was incubated with the purified human EGFR. After three rounds of panning, the dodecapeptide termed GE11 (YHWYGYTPQNVI) was shown to bind specifically to EGFR [23]. The peptide showed high binding affinity in nanomolar range (K_D_ = approximately 22 nM) that could be inhibited by EGF, indicating that they bound to the same site on EGFR. In vivo targeting ability of the radioactively labeled ^125^I-GE11 peptide was validated after intravenous (i.v.) infusion of the conjugate into nude mice bearing subcutaneous hepatocellular carcinoma (SMMC-7721) xenografts. The tumor targeting of ^125^I-GE11 was inhibited by a 100-fold excess of cold GE11 peptide [23]. In addition, GE11 peptide was used as a gene delivery vehicle upon conjugation to the polyethylenimine (PEI) polyplexes using a glycine (G-G-G-S) linker system and intravenously administered to the SMMC-7721 bearing mice [23].

Cheng and colleagues reported the conjugation of GE11 to Doxil for targeted in vivo delivery to EGFR-overexpressing non-small cell lung cancer (A549) [196]. Doxil (PEGylated liposomal Dox, PEG-LP/Dox, or non-PEGylated Dox, LP/Dox) is an FDA approved chemotherapy comprising doxorubicin drug encapsulated in liposomes. Doxil, also referred to as Lipo-Dox, is widely used in the clinics for ovarian cancer treatment [197,198].

Very promising results were obtained when the GE11-conjugated doxorubicin filled liposomes (GE11-PS-Dox) were used to treat ovarian cancer xenograft bearing mice, indicating GE11-guided drug targeting as an alternative for treatment of EGFR overexpressing ovarian cancers [199]. GE11-PS-Dox treatment very significantly increased the survival rate with minimal toxicity compared to non-targeting control (PS-Dox) or the clinically used Lipo-Dox [199]. In a similar fashion, GE11-PS-Dox also showed prolonged survival compared to the PBS- and non-targeting treatment controls (PS-Dox, Lipo-Dox) in a preclinical liver cancer treatment study [200]. Moreover, several GE11-targeted radionuclides have been studied in imaging of EGFR overexpressing tumors with either PET [201,202,203,204] or SPECT [205,206].

#### 2.3.2. MMP2/9-Targeting Peptide-CTT

Matrix metalloproteinase 2 (MMP2 or gelatinase A), MMP9 (gelatinase B) and other MMPs including collagenases, stromelysins, and the membrane-type MMPs belong to a large family of endopeptidases involved in the degradation of extracellular matrix (ECM) proteins constituting the basement membrane barrier as well as cleavage of growth factor precursors, receptor tyrosine kinases, cell adhesion molecules, and other proteins [207,208]. MMPs are involved in tumor growth, migration, invasion, and metastases as well as angiogenesis and tissue remodeling [207]. Interestingly, MMP9 expression in the stromal cells but not in the breast cancer cells associated with poor prognosis [209]. In an attempt to develop peptide-based gelatinase inhibitors to attenuate cancer progression, an in vitro phage display screen with purified MMP9 using a random peptide (CX_5_-_8_C) library was used to identify two cyclic decapeptides (CTTHWGFTLC (CTT) and CRRHWGFEFC) targeting MMP2 and MMP9 [24]. These peptides share the HWGF domain. The inhibitory effect of the peptides was analyzed in a ^125^I-gelatin degradation assay. CTT or CRRHWGFEFC inhibited the MMP9 activity with micromolar IC_50_ values while the control peptide (GACLRSGRGCGA) showed negligible effect. CTT also inhibited MMP2 with IC_50_ of 10 µM [24]. CTT inhibits specifically the MMP2 and MMP9 activity while no inhibitory activity against MT1-MMP, MM8 or MM13 was detected [24]. As a further modification, a CTT derived retro inverso peptide (D amino acids except glycine, L-Gly) was generated (cltfGwhttc) and it showed better inhibitory effect against MMP2 than the original CTT in vitro. However, it was not studied further due to its poor water solubility [25].

When the gelatinase targeting CTT peptide was developed for radioimaging, it lost the inhibitory activity after conjugation to ^125^I, but not when conjugated to Technetium-99m [25]. Thus, the N-terminus of CTT was modified by the addition of two alanine and one tyrosine residues (AAY-CTT) followed by labeling with ^125^I (^125^I-AAY-CTT) to restore the inhibitory activity. The ^125^I-AAY-CTT as well as the CTT coated ^125^I-BSA encapsulated liposomes accumulated to the KS1767 Kaposi’s sarcoma tumors following the intravenous injections, unlike their non-targeting controls with some indication of homing also to metastatic lesions in the lung [25].

In another study, CTT peptide conjugated to ^64^Cu through a DOPA (1,4,7,10-Tetraazacyclododecane-1,4,7,10) chelator was tested in PET imaging. Even though the Cu (II)-DOTA-CTT inhibited MMP2 and MMP9 activities with binding affinities (EC50) of 8.7 µM and 18.2 µM, respectively, which are very similar to those of the original CTT (13.2 µM and 11.0 µM, respectively), it was not successful in in vivo tumor imaging [210]. Stability and the gelatinase inhibition activity of the CTT peptide was increased by substituting the disulfide bond with an amide bond to form a c(KAHWGFTLD)NH_2_ peptide (C6). Cy5.5 fluorescein conjugated to the C6 peptide was taken up by MMP-2 expressing glioma cells in vitro and homed to both intratibial PC-3 prostate xenografts and orthotopic U87 glioma xenografts in vivo [26]. In addition, C6 peptide conjugated to NOTA (1,4,7,10-Triazacyclononanetriacetic acid) chelator and radiolabeled with ^68^Ga (^68^Ga-NOTA-C6) showed accumulation of the conjugate in subcutaneous ovarian cancer (SKOV3) xenografts [211]. Several studies have since utilized CTT as an MMP2 inhibitor as vasorelaxant [212] and imaging of gelatinase activity in tumors [213].

#### 2.3.3. Grb7 Adapter Targeting Cyclic Peptide

Human growth receptor bound protein 7 (Grb7) is a 532 amino acid long adapter protein implicated in cancer progression and invasion upon interaction with its binding partners comprising various receptor tyrosine kinases (RTKs) such as epidermal growth factor receptor (ErbB2/HER2) [214], ErbB3 and ErbB4 [215], focal adhesion kinase (FAK) [216], and platelet-derived growth factor (PDGF) receptors [217]. The overexpression of these RTKs in many cancer types including breast, gastric, and esophageal cancers have been shown to promote their invasive activity [218]. The C-terminal Src homology (SH2) motif of Grb7 is required for the upstream binding events leading to the activated signaling pathways. To develop inhibitors specific for the Grb7 SH2 domain, a phage display screen on purified Grb7-SH2 domain was designed. First, a screen with a phage library, which displays 18 random amino acids with two cysteines fixed at positions 5 and 16 (X_4_-Cys-X_10_-Cys-X_4_) was performed. Based on the enriched sequences in the first screen and the fact that Grb7 binds its cellular targets via the phosphorylated tyrosine residue in RTKs (pYXN), a YXN-containing phage library was designed (X_4_-Cys-X_4_-Tyr^10^-(Asp/Ala/Glu/X)^11^-Asn^12^-X_3_-Cys-X_4_) and used for the subsequent screening. After further validations and modifications, a novel non-phosphorylated cyclic peptide (WFEGYDNTFPC = G7-18NATE), specifically targeting the SH2 domain of Grb7 (Grb7-SH2) was discovered [27]. No binding was observed to the other Grb-family members sharing similar amino acid sequence such as Grb14-SH2 (67% similarity) or Grb2-SH2 [27]. G7-18NATE inhibited the interaction between Grp7 and ErbB family RTKs, in particular ErbB3, and the cyclic structure was required for the inhibitory activity [27]. G7-18NATE has been further optimized following modification with cell penetrating peptides (CPPs) to increase the cellular uptake. G7-18NATE conjugated to two different CPPs (TAT or penetratin) showed enhanced cellular uptake and inhibitory effect in vitro in metastatic breast cancer cells such as SK-BR-3, ZR-75-30, MDA-MB-231, and MDA-MB-361 compared to the non-malignant breast cancer cells such as MCF7, MCF10A, mouse fibroblast 3T3 cells or non-targeting control peptides (free TAT, penetratin, negative control TAT, negative control penetratin) treated cells [219]. However, no in vivo studies have been published and thus, the efficacy of the peptide as an anti-tumor agent remains to be studied.

### 2.4. Tumor-Associated Macrophage Binding Peptide—UNO

Tumor-associated macrophages (TAMs) have been linked to the tumor progression and suppress anti-tumor immunity. Number of TAMs in human tumors correlate with a higher tumor grade and shortened survival for several cancer types [220]. In particular, the perivascular macrophages in tumors have been associated with increased tumor angiogenesis, distant metastasis, poor prognosis, and/or the recurrence of tumors after chemotherapy in various forms of cancer [221]. To identify peptides binding the peritoneal cells of 4T1 breast cancer bearing mice, Teesalu and colleagues performed a phage screen by injecting the CX_7_C peptide library to the peritoneum of tumor-bearing or healthy mice [28]. After 2 h, the peritoneal cells were isolated, the bound phage were rescued by amplification and subjected to high-throughput sequencing. The CSPGAKVRC peptide (UNO) was the most frequent non-truncated peptide in the pool amplified from the tumor-bearing mice. The GSPGAK motif was detected in physiological ligands of CD206, a marker of M2-skewed macrophages [28]. The authors then showed that UNO bound to the CD206, but interestingly the binding required linearity of the UNO peptide. It was concluded that the glutaredoxin system in the tumor microenvironment may contribute to the reduction of the disulfide-bridged UNO. Intravenously injected fluorescent UNO (FAM-UNO) accumulated in the CD206+ macrophages in 4T1 tumors and in sentinel lymph nodes. In addition, FAM-UNO recognized the CD206+ macrophages in several other tumor models such as intracranial glioblastoma (WT-GBM), metastatic melanoma (B16F10), and peritoneal gastric carcinoma (MKN45-P). Moreover, FAM-UNO was shown to guide cargo to CD206+ macrophages and was proposed as a potential sentinel lymph node imaging agent [28].

## 3. Cell Penetrating Peptides

Due to the poor or non-permeable nature of some tumor targeting peptides and their associated cargoes across cell membranes, it has become essential to modify the permeability of these peptides. One way to increase the permeability is the conjugation to known cell penetrating peptides (CPPs). CPPs are short (5–40 amino acids long) peptides with the ability to gain access to the cell interior by means of different mechanisms and with the capacity to promote the intracellular delivery of covalently or non-covalently conjugated cargoes [222]. CPPs can enter any type of cells and therefore lack ability to target specific cell types. The widely studied CPPs that derive from natural proteins include the 12-mer human immunodeficiency virus (HIV) derived trans-activator of transcription peptide (TAT; ^48^GRKKRRQRRRPPQ^60^) [29] and a 16-mer *Drosophila* Antennapedia derived peptide called penetratin (P16; ^43^RQIKIWFQNRRMKWKK^58^) [30]. The different mechanism of internalization of CPPs and other targeting peptides alike are not within the scope of this review.

### 3.1. TAT

TAT (48–60), GRKKRRQRRRPPQ, is an arginine rich peptide that can cross the cellular membranes and translocate into the nucleus due to a nuclear localization signal within its sequence [223]. TAT has been conjugated to various active agents to mediate their delivery into the cells [224]. Tan et al. utilized TAT-derived peptides (P1, P2, P3) coupled to an anti-HER2 peptide mimetic (AHNP) and STAT3 inhibiting peptide (P1/P2/P3-AHNP-STAT3) to enhance the TAT-dependent internalization and delivery of the therapeutic peptide into different HER2-overexpressing breast cancer cells including MDA-MB-435, SKBr3, and BT-474. The P1 variant (YGRKKRRQRRR) showed highest internalization compared to the other two truncated variants (P2; YGRKKRRQRR) and (P3; YGRKKRRQR) in HER 2-overexpressing SKBr3 cells indicating that arginine residue (R) is important for the increased cellular internalization [225].

### 3.2. Penetratin

Penetratin, like TAT, crosses the biological membranes and is transported to the nucleus of the cells [30]. The interaction between penetratin (P16; ^43^RQIKI**W**FQNRRMK**W**KK^58^) and phospholipid bicelles mimicking biological membranes was studied by NMR spectroscopy. The secondary structure and positioning studies confirmed the two tryptophan residues (^48^W and ^56^W) as the most important amino acids required for cellular internalization [226,227]. The P16 or its shortened variant P7 (RRMKWKK) has been utilized to promote cellular uptake of impermeable targeting peptides such as G7-18NATE [228]. Antitumor efficacy and translocation across the BBB of liposomes or liposome encapsulated drugs (doxorubicin and erlotinib) were evaluated using the in vitro BBB and intracranial glioblastoma xenograft models [229]. Dual conjugation of penetratin and transferrin to liposomes (Tf-Pen-Lipo) significantly enhanced the BBB translocation. In addition, doxorubicin/erlotinib containing Tf-Pen-Liposomes very significantly accumulated in tumors, inhibited tumor growth, and prolonged survival rate when compared to the free drugs [229].

### 3.3. Cytotoxic CPP

A novel cytotoxic CPP derived from the tumor suppressor p14ARF protein was identified [11]. The 22 amino acid CPP known as ARF (1-22) internalized efficiently and inhibited the growth of MCF7 and MDA-MB-231 breast cancer cells via the induction of apoptosis [11]. Our lab has also taken advantage of the cytotoxic CPP to enhance internalization of the CooP peptide conjugated to chlorambucil for treatment of mice bearing intracranial glioma xenografts [8].

## 4. Tumor Penetrating Peptides

Poor penetration of active agents into the tumor parenchyma is one of the limiting factors of tumor-targeted therapy or imaging. The discovery of the tumor penetrating peptides aids the development of more efficient anti-tumor therapies. LyP-1 was the first tumor penetrating peptide identified. It was very rapidly distributed throughout the tumor after intravenous injection [22]. However, at the time of its discovery the mechanism of tumor penetration was not understood. It was the discovery of the iRGD peptide that shed light to the pathway utilized by the tumor penetrating peptides [31,230]. The tumor penetrating peptides contain the CendR recognition domain responsible for the neuropilin-1 dependent internalization [231].

### 4.1. iRGD

Recently, three related RGD containing variants (CRGDKGPDC, CRGDRGPDC, and CRGDKGPEC) were identified by an ex vivo/in vivo phage display screen [31]. The CRGDKGPDC peptide showed the highest internalization into human prostate cancer (PPC1) cells in vitro and was named iRGD [31]. The introduction of an extra cysteine (Cys) residue increased both the half-life and tumor accumulation of the iRGD (Cys-iRGD) [232]. Tumor penetrating peptides share a R/K/XXR/K internalization motif that cause vascular leakage and 3-step tissue penetration [31,230,231]. iRGD peptide first accumulates at the surface of the αv-integrin expressing endothelial and other cells in tumors. The RGD peptide is then proteolytically cleaved to expose the cryptic C-terminal CendR (C-end rule) element RXXK/R. The CendR domain then mediates the binding to another cell surface receptor, neuropilin-1 (NRP-1). The latter interaction triggers the internalization of iRGD and its associated cargo through a “CendR” internalization pathway into the extravascular tumor parenchyma [31,230,231]. The fluorescence-labeled iRGD (FAM-iRGD) showed increased fluorescence accumulation and NRP-1 colocalization in the cultured PPC1 cells compared to the controls; the non-CendR domain containing peptide (CRGD**G**GPDC) and convectional RGD (CRGDC and RGD-4C). Importantly, the in vivo data confirmed iRGD peptide penetration and extravasation via binding to αv integrins and showed NRP-1 colocalization after the i.v. infusion of the FAM-iRGD in ductal adenocarcinoma (PDAC) tumor bearing mice compared to the controls indicating NRP-1 dependent cellular internalization of the iRGD via the CendR pathway [31]. In addition, iRGD-abraxane (a 130 nm nanoparticle consisting of albumin-embedded paclitaxel) treatment was superior compared to free abraxane or CRDGC-abraxane in inhibiting the growth of orthotopic human prostate (22Rv1) and breast cancer (BT474) models [31]. iRGD peptide has also shown to enhance delivery of co-administered drugs [233,234].

iRGD has been used in a wide array of cancer models for specific targeting in conjugation with chemotherapy and nanoparticles and other active agents. In a recent study, a peritumorally injected pore-forming alginate gel loaded with granulocyte-macrophage colony-stimulating factor (GM-CSF) and doxorubicin-iRGD (Dox-iRGD) conjugate significantly improved the antitumor efficacy against poorly immunogenic triple negative breast cancer (4T1) model [235]. iRGD incorporated into a self-assembling prodrug polymer delivery system consisting of nanosized polymeric micelles, camptothecin, and photosensitizer IR780 (CPT-S-S-PEG-iRGD@IR780) for combination therapy showed anticancer activity after crossing the BBB in in vivo orthotopic glioma model [236].

### 4.2. iNGR

Using the iRGD template (CRGDK/RGPDC), Alberici and colleagues synthesized a tissue-penetrating variant of the NGR peptide, which binds to the CD13 on endothelial cells [16,18,171] by adding the CendR internalizing domain to generate a novel peptide designated as iNGR (CRNGRGPDC) [19]. Incorporation of the CendR sequence enhanced the homing, penetration, and anticancer activity of iNGR when co-administered with doxorubicin in 4T1 breast cancer xenograft bearing mice [19]. Kang and colleagues utilized the iNGR peptide for targeted delivery and penetration of chemotherapy (paclitaxel) encapsulated nanoparticles (PEG-PLGA) to intracranial glioblastomas and were able to prolong the survival of tumor-bearing mice [237].

### 4.3. tLyP-1

The LyP-1 sequence (CGNKRTRGC) contains a cryptic CendR domain responsible for the tissue penetrating activity [22]. The truncated variant of the LyP-1 peptide, tLyP-1 (CGNKRTR), with exposed CendR domain showed more potent tumor homing and penetration than the parent peptide. In addition to NRP-1, the tLyP-1 also shows affinity towards NRP-2 to trigger cellular internalization via the CendR pathway [32].

### 4.4. PL3

Another CendR motif containing peptide is the Tenascin-C binding peptide PL3 (AGRGRLVR) that was identified by using the in vitro biopanning on recombinant Tenascin-C [33]. PL3 functionalized iron oxide nanoworms (NW) or Ag-nanoparticles accumulated in the glioblastoma or pancreatic carcinoma xenografts in preclinical models and colocalized with Tenascin-C and NRP-1. In addition, treatment of glioblastoma-bearing mice with PL3-guided proapoptotic _D_(KLAKLAK)_2_-peptide containing NWs significantly inhibited tumor growth and prolonged the survival of the animals [33].

## 5. Peptides from Native Ligands

Several natural regulatory ligands including hormones have been characterized and optimized for diagnostic and/or therapeutic applications. Here, we discuss two prominent native peptides; somatostatin and alpha-melanocyte stimulating hormone (α-MSH) and their analogues (Table 2). These peptides have been exhaustively studied both in preclinical and clinical trials.

### 5.1. Somatostatin Receptor Targeting Peptide

Somatostatin (SST), a cyclic regulatory peptide hormone of 14 amino acids, originally discovered during studies with the extracts from rat and sheep hypothalamus exhibit a wide range of biological activities including the inhibition of growth hormone secretion and inhibition of cellular growth via the induction of apoptosis [242,243,244]. Human somatostatin receptors are transmembrane G-protein coupled receptors consisting of five different subtypes (SSTR1-5) that are widely expressed in both normal tissues and solid cancers [245]. SSTRs are predominantly expressed by human central nervous system (CNS) and neuroendocrine tumors [246]. However, several cancers, including breast [247,248,249], lung [250,251], some colorectal, and ovarian cancers show high SSTR expression [252]. Despite the high affinity of native somatostatin towards SSTRs, its clinical use was hampered by the need of intravenous administration, short half-life, and post-infusion hypersecretion of hormones [253]. Thus, somatostatin analogues devoid of the somatostatin weaknesses have been developed. Octreotide (SMS 201-995), an 8-mer cyclic peptide derived from the native somatostatin, was the first clinically used somatostatin analogue. Octreotide inhibits the release of glucagon and insulin, has a half-life of over 2 h, and is over 20 times more active than the native somatostatin [34,254]. However, while the native somatostatin binds to all SSTRs, octreotide binds with high affinity to SST2 and SST5. We showed the conjugation of octreotide to cryptophycin drug through a cleavable linker (Val-Cit). The extension of the conjugate with a self-immolative sequence (Gly-Pro) improved the plasma stability. In the same study, octreotide was fluorescently conjugated to Cy5.5 dye and was efficiently internalized in the murine AtT20 pituitary tumor cell line overexpressing SSTR2 and homed to the AtT20 tumor xenografts [255]. Octreotide and other somatostatin analogues have been widely used in the clinic for imaging and treatment of different types of cancers [256].

### 5.2. Melanocortin-1 (MC1) Receptor Targeting Peptide

Melanocortin receptors (MCRs), MC1R-MC5R, belong to the GPCR super family. MC1R shows the highest expression in melanomas and in the majority of human metastatic melanoma derived tumor cells. MC1R is primarily involved in the skin pigmentation and expressed by normal human skin cells including keratinocytes and melanocytes [257,258]. Alpha-melanocyte stimulating hormone (α-MSH) (^1^SYSMEHFRWGKPV^13^) is a 13 amino acids long peptide that binds MC1R with high affinity [259]. SAR data from alanine substitution and tyrosinase activity measurement showed that Met^4^-His^6^-Phe^7^-Arg^8^-Trp^9^ are important for the binding to melanocortin receptor and biological activity [260].

The cyclization of α-MSH hexapeptide via lactam bridge (CycMSH_hex_) was utilized for improved SPECT (^203^Pb-DOTA-GGNle- CycMSH_hex_) and PET (^68^Ga-DOTA-GGNle- CycMSH_hex_) imaging of melanoma using the B16/F1 and B16/F10 xenograft models [261,262]. Further modifications to α-MSH showed that replacement of Glu^5^ with Asp^5^ (E^5^-D^5^) and Gly^10^ with Lys^10^ (G^10^–K^10^), enhanced the biological activity [263]. Norleucine (Nle), an isomer of leucine, have been used to generate variants of α -MSH. The Nle^4^-D-Phe^7^-α-MSH octapeptide variants such as MSH_oct_ (βA^3^-Nle^4^-D^5^-H^6^-f^7^-R^8^-W^9^-K^10^ -NH_2_), NAPamide (Nle^4^-D^5^-H^6^-f^7^-R^8^-W^9^-G^10^-K^11^-NH_2_), and unmodified α-MSH (Ac-S^1^ -Y^2^ -S^3^ -M^4^ -E^5^ -H^6^ -F^7^ -R^8^ -W^9^ -G^10^ K^11^ -P^12^ -V^13^) conjugated to DOTA chelator and radiolabeled with ^67/68^Ga or ^111^In have been evaluated for the PET imaging of melanoma (B16F1) bearing mice. DOTA-NAPamide showed superior binding affinity and in vivo biodistribution when labeled with ^67^Ga, ^68^Ga or ^111^In compared to DOPA- MSH_oct_ and control, DOPA- MSH [35].

Several other studies utilizing Nle^4^ -D-Phe^7^-α-MSH derived octapeptide or its cyclized analogue to enhance radionuclide imaging of melanoma or metastatic melanoma lesions have been documented [264,265,266,267].

The Nle^4^ -D-Phe^7^-α-MSH derived octapeptide (Ac-[NIe^4^, D-Phe^7^]-α-MSH_4-11_ -NH_2_) have been utilized as MC1R antagonist after showing overwhelmingly greater biological activity and plasma stability than the unmodified α-MSH in several biological assays including adenylate cyclase, tyrosinase activity, and frog skin assays [268,269,270]. The analogue designated as afamelanotide is widely used in many clinical trials for the treatment of several skin related diseases such as erythropoietic protoporphyria [271,272,273,274], vitiligo [275], Hailey-Hailey disease [276], acne vulgaris [277], solar urticaria [278], and eumelanogenesis [279]. A more recent strategy using photosensitizer moiety attached to α-MSH for targeted therapy have been adopted [280,281].

## 6. Therapeutic Peptides

Several therapeutic peptides in addition to peptide-based therapeutics have evolved over the years for preclinical applications. However, the successful translation of therapeutic peptides for clinical applications would rely on enhanced tumor targeting with simple and optimized conjugation chemistry that would not interfere with the binding affinities. Here, we explored several conjugation strategies for therapeutic peptides in different cancer models. These anti-cancer toxins are mostly naturally derived bioactive peptides from animal venoms. For example, (KLAKLAK)_2_ [238], melittin [282], mastoparan [239], and its derivative mitoparan (MitP) [240] were derived from insect venoms [283]. Marine organisms also provide an excellent source of potent anticancer agents including monomethyl auristatin E (MMAE), which has been developed further, e.g., as drug antibody conjugate for cancer therapy [284].

(KLAKLAK)_2_ (referred to as KLAK from here) is an antimicrobial peptide known to induce apoptosis via mitochondrial membrane disruption [285]. KLAK exerts low cytotoxicity to mammalian cells owing to its low internalization capacity and thus requires an internalizing peptide for effective anticancer activity. KLAK has been conjugated to several peptides such as RGD-4C and NGR [127], NRP-1 binding peptide (CGFYWLRSC) [286], and bladder tumor homing nonapeptide (Bld-1; CSNRDARRC) [287]. Conjugation strategy that utilizes the coupling of pro-apoptotic KLAK peptide either to the C- or the N-terminus of the targeting peptide have been used. Most of the peptides identified using the T7 phage display system use the N-terminus for conjugation. However, it is important not to interfere with the binding site of the targeting peptide while considering the conjugation strategy. The iron oxide nanoworms were functionalized by conjugation of the D(KLAKLAK)_2_ to the N-terminus of the p32 targeting tumor penetrating peptide (LinTT1; AKRGARSTA [187] through a polyethylene glycol (PEG) linker (NW-PEG-Cys-D(KLAKLAK)2-X-LinTT1). The conjugate showed significantly improved anticancer activity in several p32-expressing peritoneal tumor models [288]. Modification of the N-terminus of glioma homing peptide (CooP; CGLSGLGVA [8]) by introduction of alanine and TAMRA fluorophore (TAMRA-A-CooP) did not affect CooP homing in vivo using glioblastoma model [12].

MMAE is a very potent peptide toxin and an FDA approved component of an anti-CD30 antibody drug conjugate [241]. These toxins have also been exploited for peptide-directed therapy. For instance, MMAE was similarly loaded into polypeptide micelles functionalized with the RGD peptide (MMAE-cRGD-Lipep-Ms) for enhanced targeted therapy to the HCT-116 colorectal tumor xenografts [289].

## 7. Strategies to Improve Peptide Synthesis and Their Pharmacokinetics

The success of any targeting peptide would obviously rely on an efficient and optimal synthesis strategy that does not compromise their potency. The solid-phase peptide synthesis (SPPS), originally developed by Merrifield, is the most frequently used method for peptide synthesis [290]. The method utilizes an insoluble solid polymer-based resin support, where iterative cycles of coupling and deprotection are performed (Figure 4), a concept that is lacking in the classical solution-phase peptide synthesis [291]. SPPS utilizes two different types of N^α^-protecting groups; 9-fluorenylmethoxycarbony-group (Fmoc) or tert-butyloxycarbonyl-group (Boc) as building backbone to assemble amino acids onto the resin.

We used the Rink Amide MBHA resin to synthesize the glioma targeting peptide, CooP using an Fmoc/*tert*-butyl (*t*Bu) strategy. The Fmoc protecting group was removed using piperidine in DMF (Fmoc deprotection step), followed by the sequential amino acid coupling using *N, N*–dicyclohexylcarbodiimide (DIC) and ethyl (hydroxyimino) cyanoacetate (Oxyma) as coupling reagents. Every coupling cycle was followed by the Fmoc deprotection and resin wash using DMF and CH_2_Cl_2_ to remove the excess reactants and by-products. The corresponding crude peptide was cleaved with trifluoroacetic acid (TFA) and DTT, H_2_O and TIS as scavenger agents, precipitated with cold ether, centrifuged, and purified by reversed phase HPLC [12].

The Fmoc strategy is widely used and allows the preparation of broad range of peptides (e.g., phosphorylated and glycosylated peptides) that were previously challenging with the Boc synthesis strategy [292,293]. However, the choice of synthesis strategy would depend on several factors including the composition of amino acid sequence. For instance, an aspartate containing peptides may cause aspartimide formation, a serious side chain reaction, after repeated exposure to bases like piperidine thereby giving rise to unwanted by-products during the Fmoc-based synthesis. Many researchers have utilized different approaches to overcome aspartimide formation by incorporating different protecting groups, such as dimethoxybenzyl (Dmb) [294] and 2-hydroxy-4-methoxy-benzyl (Hmb) [295] in the preparation of aspartic acid containing peptide sequences.

Another important factor limiting the success of synthetic peptide is their short circulation time often caused by proteolytic degradation, thus leading to their rapid systemic clearance. Several enzymatic stability studies have been developed using serum, liver, or kidney extracts or in vivo models to predict the targeting peptide degradation profiles [296,297,298,299,300,301]. Strategies to improve the overall pharmacokinetic profile of such peptides have been widely studied as several novel peptides have emerged after successful and improved modification of the original peptide. For instance, octreotide is a truncated analogue of the native somatostatin with substitution of L-amino acids with their D-counterparts, now endowed with longer lasting plasma stability and potency [34]. Cyclization is also an important strategy used to decrease proteolytic degradation [302]. A cyclized derivative of native α-MSH (α-melanocyte stimulating hormone), a hormone peptide against malignant melanoma, showed improved plasma stability compared to the native peptide when used as radionuclide imaging agent [303]. The N- and C-terminal modifications of MART-1 (^27^AAGIGILTV^35^), a peptide vaccine against melanoma, have also shown improved plasma stability and immunogenicity compared to the unmodified peptide [304]. It’s N-terminal glutamic acid containing variant, ELA (ELAGIGILTV) has shown prolonged stability and activity [305,306]. Although peptide vaccines are not within the scope of this review, it is worthy to mention that MART-1/ELA were in phase I/II clinical trials with vast numbers of publications. Indeed, modifications of peptides including cyclization, N-terminal acetylation, C-terminal amidation, and PEGylation, have proven to be viable strategies for improving targeting peptides’ activities.

## 8. Conclusions

For effective clinical applications of peptide-drug conjugates or other peptide-based agents, tumor-targeting peptides need to display high on-target affinity, permeability, plasma stability, and retention. Many peptides identified using combinatorial screens or derived from native proteins are usually subjected to further modifications to improve their pharmacokinetics. Hence, after the target is known, it is essential to perform structure-activity relationship (SAR) analyses to characterize the active binding domains (Figure 5). Upon target identification and validation, SAR-based assays, such as site-directed mutagenesis and affinity studies, are essential for successful development of viable tumor targeting peptides. Identification of the active binding domains provides important information for the design and development of peptide-drug conjugates. In an optimal situation, simple conjugation chemistry between a peptide and a drug without interference with the peptide’s binding domain or the drug’s efficacy would allow the development of peptide-drug conjugates, but is often hard to implement in practice for successful tumor targeting.

The wider exploitation or clinical translation of tumor targeting peptides remain challenging for several reasons. Some peptides identified using phage display screens bind poorly to their target with micromolar or even lower affinities, since the multivalent display on the phage allows compensation of low affinity with high avidity. Moreover, traditionally, we look for the enrichment of peptide sequences after the screens. However, if the increased binding of a phage pool is not due to a single peptide, but many different peptides with low affinity to the target, no enrichment of peptide sequences can be detected. In addition, some peptides show affinity that is not related to target, hence classified as target unrelated peptides (TUPs). TUPs may enrich, due to selection or propagation, and during each screen. Therefore, this should be taken into account when evaluating the binding sequences from the phage display screen. Many peptides are susceptible to proteolysis. To overcome this challenge, the N- and C-terminals are often capped by acetylation, amidation, PEGylation, cyclization, or amino acids are replaced with the D-isomer counterparts to decrease enzymatic degradation. SAR data are important information for successful peptide modification as targeting peptides may lose their potency if such modification interferes with their active binding domains.

To date, there has been a decline in the discovery of novel targeting peptides, especially from combinatorial screens, while many novel modifications of known targeting peptides have emerged as judged by the number of published reports. Whilst improvements of the original targeting peptides are important, scientists should also take advantage of the numerous peptide library techniques to screen for new peptides, using the relevant tumor models. This will open new avenues for tumor targeting peptides as therapeutic or imaging agents.

## Figures and Tables

**Figure 1 pharmaceutics-13-00481-f001:**
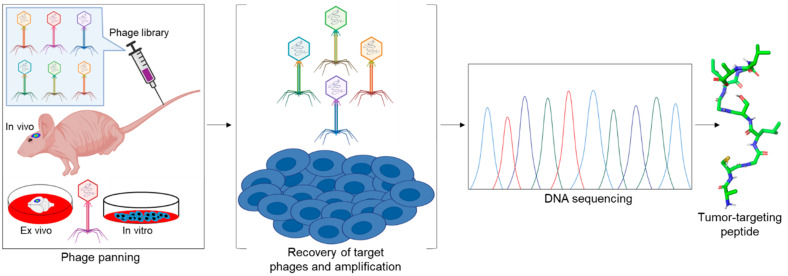
Schematic representation of phage display. In vivo, ex vivo, and in vitro biopanning of phage displayed peptide libraries, recovery of target phage, phage amplification, sequencing, and identification of tumor-targeting peptides. The figure was created with BioRender.com (access on 28 February 2021). The tumor-targeting peptide was generated from an unpublished data using discovery studio visualizer v19.1.0.18287 (BIOVIA).

**Figure 2 pharmaceutics-13-00481-f002:**
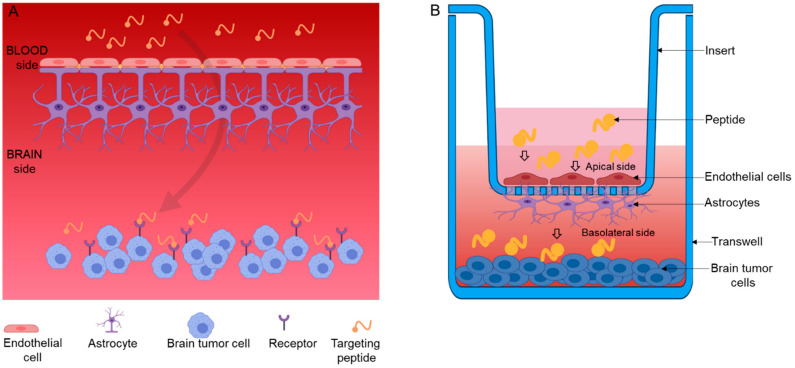
Schematic illustration of targeting peptide transport across the blood–brain–barrier (BBB) in both (**A**) in vivo and (**B**) in vitro models. Figure was created with BioRender.com. (Access on 28 February 2021).

**Figure 3 pharmaceutics-13-00481-f003:**
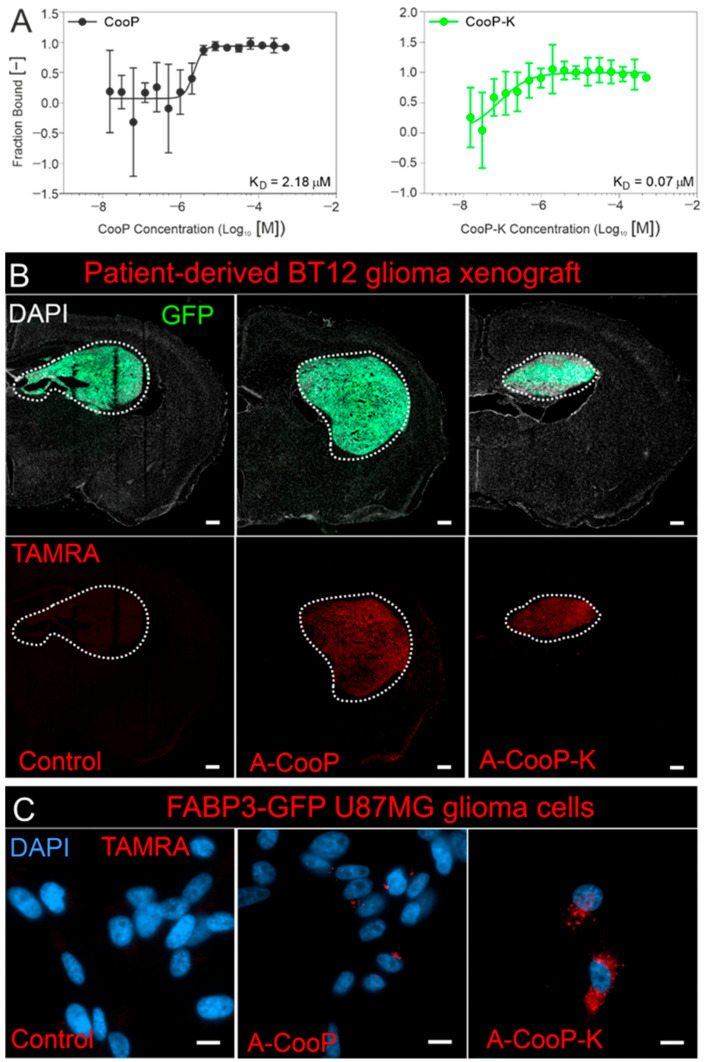
Homing and internalization of the A-CooP and A-CooP-K peptides to intracranial glioblastomas. (**A**) Affinity measurements (calculated K_D_) of the A-CooP and A-CooP-K peptides binding to recombinant FABP3/MDGI. (**B**) A-CooP and A-CooP-K home efficiently to the patient-derived BT12 glioblastoma xenografts while the control peptide showed no homing, scale bar = 1 mm. (**C**) Widefield fluorescence images of mammary-derived growth inhibitor/fatty acid binding protein 3 (MDGI/FABP3) overexpressing U87MG cells show efficient internalization of the A-CooP-K peptide compared to the A-CooP (red) with minimal internalization and control peptide with no internalization, scale bar = 10 µm. Ayo *et al.* unpublished figures.

**Figure 4 pharmaceutics-13-00481-f004:**
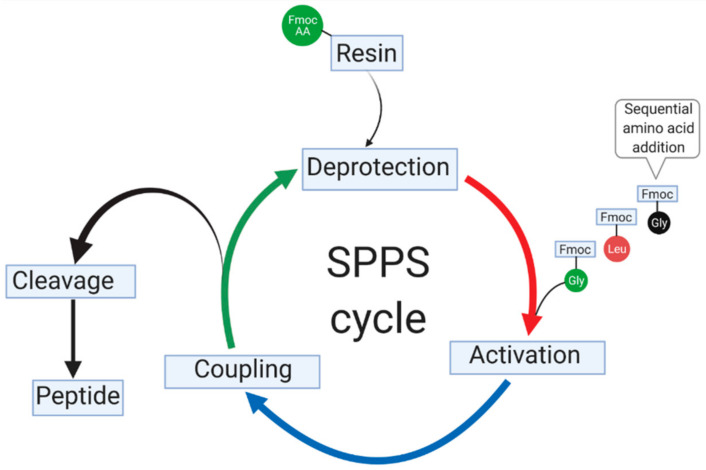
Schematic overview of repeated cycles of Fmoc-based solid-phase peptide synthesis (SPPS). The first cycle shows the attachment of the Fmoc to the resin with a side chain protecting group (top). The protecting group is removed (Fmoc deprotection step), followed by the sequential addition of each amino acid building block (activation step). The subsequent Fmoc-amino acid derivatives are coupled with the appropriate coupling reagents (coupling step). Each completed coupling cycle is followed by deprotection step before proceeding to the cleavage step and finally to the peptide purification. Figure was created with BioRender.com. (Access on 28 February 2021).

**Figure 5 pharmaceutics-13-00481-f005:**
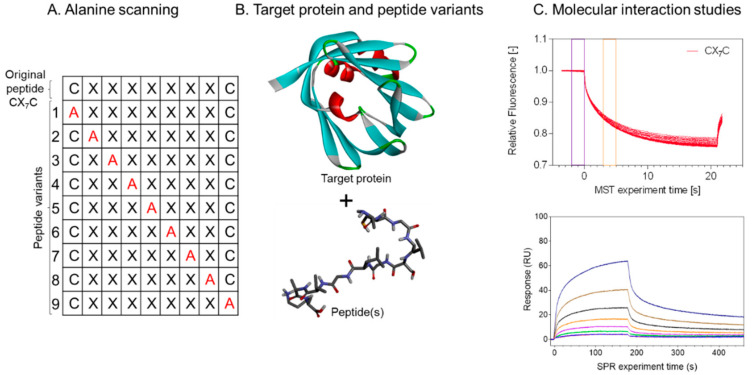
Sequence-activity relationship (SAR) analysis of a cyclic tumor targeting peptide using alanine scan and affinity measurements. (**A**) Scheme shows the sequential substitution of each amino acid residue (X) including the first and the last cysteine residues (**C**) with alanine (A in red) to generate peptide variants, 1–9. (**B**) Production of the recombinant target protein and synthesis of the original peptide and its variants (**C**) microscale thermophoresis (MST) and surface plasmon resonance (SPR) to study the molecular interactions. Figure was generated using prism 9.0.0 (GraphPad software, Inc., San Diego, CA, USA) and discovery studio visualizer v19.1.0.18287 (BIOVIA, San Diego, CA, USA) from Ayo *et al.* unpublished data

**Table 1 pharmaceutics-13-00481-t001:** Sequences of the tumor homing peptides and their target proteins, malignancies, and applications.

Target	Peptide	Sequence	Malignancy(s)	Application	Reference
FABP3/MDGI	CooP	CGLSGLGVA	Glioblastoma	SPECT/CT imaging, chemotherapy, nanoparticles	[8]
p14ARF protein	ARF (1-22)	MVRRFLVTLRIRRACGPPRVRV	Several tumor types	Cytotoxic CPP	[11]
FABP3/MDGI	ACooPK	ACGLSGLGVAK	Glioblastoma	Fluorescent imaging	[12]
Transferrin receptor (TRf)	T12	THRPPMWSPVWP	Glioblastoma, colorectal cancer	PET imaging	[13]
Low density lipoprotein receptor	Angiopep-2	TFFYGGSRGKRNNFKTEEY	Brain tumor/brain metastasis	Chemotherapy, nanoparticles, PET imaging	[14]
Interleukin-4 Receptor (IL-4R)	AP	CRKRLDRNC	Glioblastoma, lung cancer	Chemotherapy, nanoparticles	[15]
Integrins (αvβ3, αvβ5 and αvβ1)	RGD	CRGDC	Melanoma, glioblastoma, ovarian cancer	PET imaging, SPECT/CT imaging, chemotherapy,	[16]
	RGD-4C	CDCRGDCFC	Lung cancer, colorectal cancer, neuroblastoma, fibrosarcoma	Cytokine therapy, chemotherapy,	[17]
CD13/APN	NGR	CNGRC	Lymphoma, melanoma, fibrosarcoma	Chemotherapy, cytokine therapy	[18]
	iNGR	CRNGRGPDC	Breast cancer, glioblastoma	Chemotherapy, nanoparticles	[19]
Integrins (αvβ3, αvβ5 and αvβ1)	cyclicRGD	c(RGDX*_i_*)_n_	Glioblastoma, breast cancer, prostate cancer, NSCLS, ovarian cancer	PET imaging, SPECT imaging, chemotherapy,	[20]
	Cilengitide	c(RGDfN(Me)V)	Glioblastoma	Chemotherapy, radiotherapy,	[21]
p32 /gC1qR	LyP-1	CGNKRTRGC	Osteosarcomas, prostate, breast cancers,	Chemotherapy, nanoparticles	[22]
EGFR	GE11	YHWYGYTPQNVI	Hepatocellular carcinoma,	SPECT imaging	[23]
MMP2/MMP9	CTT	CTTHWGFTLC	Kaposi’s sarcoma	SPECT imaging, nanoparticles	[24]
	D-retroCTT	cltfGwhttc	Breast cancer, melanoma	ND	[25]
	C6	c(KAHWGFTLD)	Glioblastoma, prostate cancer, ovarian cancer	PET imaging, fluorescent imaging	[26]
Grb7	G7-18NATE	WFEGYDNTFPC	No in vivo data	ND	[27]
CD206	UNO	CSPGAKVRC	Tumor-associated macrophages; breast cancer, glioblastoma, metastatic melanoma, peritoneal gastric cancer	Nanoparticle, chemotherapy, fluorescent agent	[28]
HIV-1 TAT	TAT	^46^CYGRKKRRQRRR^57^	Several tumor types	Cell penetrating peptide (CPP)	[29]
Drosophila Antennapedia	Penetratin	^43^RQIKIWFQNRRMKWKK^58^	Several tumor types	CPP	[30]
Integrins (αvβ3, αvβ5 and αvβ1)	iRGD	CRGDKGPDC	Prostate cancer, breast cancer, lung cancer; NSCLS	Chemotherapy, nanoparticles,	[31]
p32 /gC1qR	tLyP-1	CGNKRTR	Prostate, breast cancers, Glioblastoma	Chemotherapy, nanoparticles	[32]
Tenascin-C	PL3	AGRGRLVR	Glioblastoma, Pancreatic tumor	Nanoparticles	[33]
Somatostatin receptor 2 (SSTR2)	Octreotide	fCFwKTCT-(Ol)	Pituitary corticotrope tumor, metastatic midgut carcinoid tumor, pancreatic neuroendocrine tumor	Chemotherapy, PET imaging, SPECT imaging, fluorescent imaging	[34]
Melanocortin-1 receptor (MC1-R)	α-MSH	^1^SYSMEHFRWGKPV^13^	Melanoma	Therapeutic peptide, radionuclide imaging	
	Nle^4^-D-Phe^7^ - α-MSH_Oct_	^4^{Nle}DHfRWGK^11^	Melanoma, metastatic melanoma	Chemotherapy, PET imaging, SPECT imaging,	[35]

The small letters correspond to D-type isomers of amino acids while the capital letters correspond to L-type isomers. The red marks indicate the modification of the original peptide, c(·) = cyclized peptide, X*i* = number(s) of amino acids, c(·)_n_ = monocyclic to multicyclic peptide, ND = not determined, {Nle} = norleucine, an isomer of leucine with no single code letter. The references listed here are only from the original or modified peptide and do not capture all the malignancies and applications documented in the review.

**Table 2 pharmaceutics-13-00481-t002:** Peptides derived from native ligands.

Origin	Therapeutic Peptide	Sequence	Reference
Amphipathic a-helical domain of antimicrobial peptide	(klaklak)2	klaklakklaklak	[238]
Insect wasp venom	Mastoparan	INLKALAALAKKIL	[239]
	Mitoparan (MitP)	INLKKLAKL(Aib)KKIL	[240]
Mollusk/cyanobacterium *Symploca*	MMAE (dolastatin 10 derivative)		[241]

The small letters correspond to D-type isomers of amino acids while the capital letters correspond to L-type isomers. The references listed here are only from the original or modified peptide and do not capture all of the malignancies and applications documented in the review.

## Data Availability

Not applicable.

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
