# Peer review of "Peptide-Based Strategies for Targeted Tumor Treatment and Imaging"

_pharmaceutics, 2021, doi:10.3390/pharmaceutics13040481_

Round 1
Reviewer 1 Report
In this review, the authors outlined the potential role of peptide serving as homing moieties on cancer tumor. This is obviously a highly useful topic to the academic community considering the fact that the authors had listed out a range of important peptide sequence currently available in literature. There are some issues that should be resolved prior to acceptance.
- A small discourse on solid state peptide synthesis (FMOC, BOC etc) could be useful to reinforce the quality of the content.
- The reviewer would prefer the inclusion of more figures. While the text is highly informative, having only 3 figures through the review is rather underwhelming.
- Perhaps the authors may want to address concern pertaining to degradation and strategies involved in improving the overall circulation time.
- The conclusion (perspective) is way too short and uninformative. This must be greatly expanded to include thoughts and weakness, future trends etc in order to provide certain quick insights into peptides as targeting strategy for therapeutic purposes.
Author Response
- A small discourse on solid state peptide synthesis (FMOC, BOC etc) could be useful to reinforce the quality of the content.
Response 1: We have now added this important information in the section 7, page numbers 35 - 37 titled ‘Strategies to improve peptide synthesis and their pharmacokinetics’. We took the opportunity to also address the challenges of peptide synthesis, which was one of the concerns raised in the comment number 3.
2. The reviewer would prefer the inclusion of more figures. While the text is highly informative, having only 3 figures through the review is rather underwhelming.
Response 2: In addition to the original 3 figures, we have now added 2 new ones. One of the figures shows the schematic illustration of targeting peptide transport across the BBB, another one shows the schematic overview of the Fmoc-based solid-phase peptide synthesis. We believe these are two important figures that would enhance the quick understanding by wider audience.
3. Perhaps the authors may want to address concern pertaining to degradation and strategies involved in improving the overall circulation time.
Response 3: Like mentioned in response 1, we addressed the enzymatic degradation among other limiting factors and the strategies to improve the overall pharmacokinetic profiles of synthetic peptides. This can be found in section 7, page numbers 35 - 37 titled ‘Strategies to improve peptide synthesis and their pharmacokinetics’
4. The conclusion (perspective) is way too short and uninformative. This must be greatly expanded to include thoughts and weakness, future trends etc in order to provide certain quick insights into peptides as targeting strategy for therapeutic purposes.
Response 4: We have widened the conclusions by addressing the key components of the review by highlighting weaknesses of targeting peptide including TUPs, proteolytic degradation and our general thoughts towards opening new avenues for novel targeting peptides.
Reviewer 2 Report
The manuscript entitled “Peptide-based strategies for targeted tumor treatment and imaging” by Ayo and Laakkonen, has reviewed peptide-based tumor-specific strategies for treatment as well as diagnosis. The information is well summarized, and the topic is relevant. However, there are some concerns which if addressed, would significantly improve presentation and usefulness of this review for the readers.
Comments:
- Please confirm the statement on page 3, line 65. As per FDA’s definition: polymer composed of “40” or fewer amino acids are called peptide.
- In Table 1 and table 2, please add name of the malignancy/tumor and functional use for each of the peptide, i.e. diagnostic or treatment.
- Please add the source of the figures. Permission should be included if figures are taken from published literature.
- Based on vast amount of literature to target TfR using peptides, TfR targeting peptide section 2.2 can be elaborated.
- Majority of this article discusses glioma or brain cancer. Please review additional literature to span other relevant malignancies.
- Please include all peptides discussed in the manuscript into the tabular summary.
- Majority of citations are older than 5-6 years. Authors should review recently published literature and reduce reference to already reviewed material to increase value of this review.
Author Response
Reviewer 2
1. Please confirm the statement on page 3, line 65. As per FDA’s definition: polymer composed of “40” or fewer amino acids are called peptide.
Response 1: We thank the reviewer for this useful information. We have edited the sentence ‘often less than 40 amino acids in length’. This can be found under section 1 and page 3 (Introduction).
2. In Table 1 and table 2, please add name of the malignancy/tumor and functional use for each of the peptide, i.e. diagnostic or treatment.
Response 2: We have included all the peptides and their modified versions, malignancies and their corresponding diagnostic and therapeutic applications to the Table 1. However, we did not feel this necessary for the Table 2 presenting the therapeutic peptides and left Table 2 as it was.
3. Please add the source of the figures. Permission should be included if figures are taken from published literature.
Response 3: All figures except Figure 3 were created with BioRender.com. Figure 3 contains our own unpublished figure panels and data were generated using prism 9.0.0 (GraphPad software, Inc), discovery studio visualizer v19.1.0.18287 (BIOVIA) and PowerPoint 2013. The source of the figures has been highlighted under the ‘Acknowledgements’ in page 40.
4. Based on vast amount of literature to target TfR using peptides, TfR targeting peptide section 2.2 can be elaborated.
Response 4: Most of the published work on TfR targeting peptide concerned peptide T7, a peptide discovered through phage display screen. However, this particular peptide was later detected to be a target unrelated peptide (TUP). We decided to expand the topic on TUPs under this section as one of factors limiting the success of tumor targeting peptides. Even though we found vast number of papers on this particular peptide in addition to other similar TfR targeting peptides with only in vitro data and no in vivo validation, we decided not to mention any of those in the paper. The addition can be found under sub-section 2.1.2, page numbers 9 – 10.
5. Majority of this article discusses glioma or brain cancer. Please review additional literature to span other relevant malignancies.
Response 5: We have now added data on peptides that were missing in the original manuscript as follows: Tumor-associated macrophage binding peptide-UNO, Tenascin-C binding peptide PL3, and Melanocortin-1 receptor targeting peptide (MSH). UNO targets metastatic melanoma, peritoneal gastric carcinoma and GBM by binding to CD206, more details can be found under section 2.4, page 23 – 24. PL3 is a Tenascin-C binding peptide targeting pancreatic carcinoma including GBM with more details on section 4.4, page number 30. MSH and analogues are peptides against melanoma and metastatic melanoma. Details are under section 5.2, page numbers 32 – 33. Already in the original paper, peptides targeting other malignancies than glioblastoma were presented. However, this information can now be easier found since we have revised the Table 1 to include the targeted malignancies as requested by the reviewer. In addition, brain tumors have been very often the target of peptide-based delivery due to the difficulty for therapies to reach the brain tumors because of the blood-brain-barrier. Therefore, also e.g. RDG-peptides have been used frequently for delivery of imaging and therapeutic agents to brain tumors.
6. Please include all peptides discussed in the manuscript into the tabular summary.
Response 6: We have now included all the peptides and their modified versions, malignancies and their corresponding diagnostic, and therapeutic applications in Tables 1 and 2.
7. Majority of citations are older than 5-6 years. Authors should review recently published literature and reduce reference to already reviewed material to increase value of this review.
Response 7: We have cited older references of original work in some cases and minimized the usage of Review articles. After a new literature search, we have added more recently published articles on the newly added targeting peptides. Thus, we have increased the number of reviewed papers published during 2017-2021 from 45 in the original manuscript to 54 in this revised version. We now have a total number of 306 references. While reviewing the literature it became evident that very few new homing peptides have been discovered during recent years and most of the recent publications study the modified versions (including different cargoes) of the old homing peptides.
Round 2
Reviewer 1 Report
The authors had improved the paper substantively and should be accepted after minor grammar checks.
Reviewer 2 Report
The authors of the review have adequately addressed all concerns and have substantially improved the manuscript with relevant information. The revised manuscript is recommended for publication.